# Association between Quality of Life and Physical Functioning in a Gardening Intervention for Cancer Survivors

**DOI:** 10.3390/healthcare10081421

**Published:** 2022-07-29

**Authors:** Harsh Sharma, Vernon S. Pankratz, Wendy Demark-Wahnefried, Claire R. Pestak, Cindy K. Blair

**Affiliations:** 1Department of Internal Medicine, University of New Mexico, Albuquerque, NM 87131, USA; vpankratz@salud.unm.edu (V.S.P.); ciblair@salud.unm.edu (C.K.B.); 2University of New Mexico Comprehensive Cancer Center, Albuquerque, NM 87102, USA; clpestak@salud.unm.edu; 3Department of Nutrition Sciences, University of Alabama at Birmingham (UAB), Birmingham, AL 35294, USA; demark@uab.edu; 4O’Neal Comprehensive Cancer Center, University of Alabama at Birmingham, Birmingham, AL 35294, USA

**Keywords:** cancer survivors, gardening, vegetable, quality of life, COVID-19, physical function, diet

## Abstract

Purpose: To examine potential factors associated with maintaining or improving self-reported physical function (PF) among older cancer survivors participating in a gardening intervention impacted by the Coronavirus 2019 (COVID-19) pandemic. Methods: Thirty cancer survivors completed a home-based gardening intervention to encourage a healthier diet and a more active lifestyle. Device-based measures of physical activity (PA) and surveys to evaluate quality of life (QOL; PROMIS-57 questionnaire) were administered at baseline, mid-intervention (6 months), and post-intervention (9 months). Results: Depression, fatigue, and sleeplessness at baseline were significantly associated with worse average PF scores across follow-up (2.3 to 4.9 points lower for every decrease of 5 points in the QOL score; *p*-values < 0.02). Worsening of these QOL domains during the intervention was also associated with an additional decrease of 2.1 to 2.9 points in PF over follow-up (*p* values < 0.01). Better social participation and PA at baseline were significantly associated with better average PF scores during the intervention (2.8 to 5.2 points higher for every 5-point increase in social participation or 30 min more of PA; *p* values < 0.05). Every 5-point increase in pain at baseline, or increases in pain during the intervention, was associated with decreases of 4.9 and 3.0 points, respectively, in PF. Conclusions: Worse QOL scores before and during the intervention were significantly associated with worse PF over follow-up. Encouraging social participation and PA through interventions such as home-based gardening may improve long-term health among older cancer survivors.

## 1. Introduction

Older cancer survivors, defined as adults aged 65 and older, represent nearly 12.8 million (76%) of cancer survivors in the United States [1], and are more likely to have multiple chronic conditions and a poorer health status [2]. Older cancer survivors are at increased risk for osteoporosis and poor physical function (PF) compared to their cancer-free counterparts [3,4]. Factors such as decreased bone mass, limited exercise capacity, and restricted flexibility contribute to a loss of PF in older adults [5]. Cancer treatments, including systemic chemotherapy, can exacerbate these and other factors [6,7,8]. As such, cancer survivors are especially susceptible to the downstream effects of low PF, namely the development of additional disability, negative impact on quality of life (QOL), and ability to maintain independence. Adherence to a healthy lifestyle, such as a healthy diet and regular physical activity, can help prevent or attenuate these negative health outcomes.

Compared to individuals without cancer, cancer survivors are less likely to meet diet and physical activity recommendations. A population-based study identified that cancer survivors consume more empty calories (calories from solid fats, alcohol, and added sugars) compared to individuals without cancer and have lower than the recommended intake of fiber, vitamin D, vitamin E, potassium, and calcium [9]. In addition, less than half of cancer survivors consume at least five daily servings of fruits and vegetables [10]. Cancer survivors are also less likely to meet the recommended 150 min of moderate intensity or 75 min of strenuous-intensity PA per week [10]. Despite the number of efficacious interventions that have resulted in improvements in diet and physical activity, few interventions are available to assist cancer survivors once the research funding ends. Regular gardening activity is associated with numerous benefits, including improvements in well-being, lower levels of perceived stress, and increased PA [11].

Fruit and vegetable gardening offers individuals a simple and holistic means to improve diet quality, physical activity (PA) levels, and social health [12,13,14,15]. Pairing gardening interventions with the vulnerable older cancer survivor population can offer several sustainable benefits. For example, exposure to the natural environment, especially during the lockdowns enacted during the COVID-19 pandemic, was associated with reduced psychological distress and stress levels [16,17]. Older adults who garden report a sense of restoration and achievement; they also view gardening as a vital means to participate in regular physical activity and exercise [17].

Harvest for Health is a home-based vegetable gardening intervention designed to encourage a more active lifestyle and healthier diet, and improve QOL among cancer survivors [12,18,19]. Harvest for Health was initially developed and tested in Alabama [12,18,19,20]. Findings suggest that tending a home-based vegetable garden can result in meaningful increases in vegetable intake, PA, and improvements in QOL [18,20], especially physical function. We adapted the intervention to suit the different climate and growing conditions in New Mexico [21,22]. The Southwest Harvest for Health feasibility study was launched in February 2020, concurrent with the emergence of COVID-19 in the United States. Thirty participants completed the 9-month gardening intervention in late 2020 [21]. Noteworthy results include high recruitment, retention, and satisfaction rates, as well as an increased consumption of fruit and vegetable servings per day. However, changes in PA and self-reported PF were highly variable, warranting further study. Therefore, in the context of a home-based gardening intervention that occurred during the COVID-19 pandemic, this study sought to identify factors associated with changes in self-reported PF among older cancer survivors.

## 2. Materials and Methods

The Southwest Harvest for Health research protocol was published previously [22]. Briefly, a single-arm feasibility study was conducted with thirty cancer survivors who were individually mentored by Master Gardeners. Participants underwent assessments at baseline, and at mid-intervention (6 months) and post-intervention (9 months). While the study was adapted and initiated prior to the COVID-19 pandemic (baseline assessment in February 2020), the gardening intervention and follow-up assessments occurred during the pandemic (March through November 2020). All participants provided written informed consent before baseline assessment. The study was approved by the University of New Mexico Health Sciences Center (UNM HSC) Institutional Review Board.

### 2.1. Study Participants

Thirty older cancer survivors (aged 50 years and older) were enrolled and completed the intervention (100% retention). Recruitment efforts included flier distribution at community centers and referrals from oncologists and nurse navigators. Interested individuals were screened for eligibility to ensure they: (1) had a current or prior diagnosis of cancer (any type, any stage at diagnosis; physician approval for individuals with metastatic disease); (2) resided on property capable of accommodating a 1.2 m × 2.4 m raised garden bed or four (62.2 cm × 52.1 cm) garden containers; (3) had access to running water outdoors; (4) could participate in the 9-month intervention; and (5) were healthy enough to engage in daily light PA (e.g., bending, kneeling, able to walk three blocks). Among pertinent exclusion criteria were: (1) already meeting recommended guidelines for fruit and vegetable intake (>5 servings/day); and (2) already meeting recommended guidelines for PA (>150 min per week of moderate-to-vigorous intensity PA); (3) had recent experience with vegetable gardening (past year).

### 2.2. Gardening Intervention

During the intervention, cancer survivors received mentoring on establishing three seasonal home gardens from Master Gardeners, who are part of the Cooperative Extension System. The Cooperative Extension System is a U.S. Department of Agriculture program aiming to disseminate learned knowledge and benefit people in rural and urban settings [23]. Master Gardeners are volunteers who provide research-based horticulture education to communities nationwide [24]. The participant and Master Gardener dyads worked together to create and maintain three seasonal gardens (spring, summer, and fall). Key materials were distributed to participants, including gardening supplies, plants, and seeds. Due to the COVID-19 pandemic, all monthly home visits by the Master Gardeners were replaced with an extra email or telephone call.

### 2.3. Outcomes and Measures:

#### Sociodemographic and Health Data

Data were collected for participant characteristics such as age, sex, race, education level, and living arrangement (alone or with others). Health data collected included self-reported general health, number of comorbidities/chronic conditions, and cancer type. Baseline assessments were conducted during a home visit. Six- and nine-month follow-up assessments were conducted remotely (e.g., via digital or paper survey) given the pandemic restrictions.

The outcome of interest was self-reported PF. The Patient Reported Outcomes Measurement Information System (PROMIS)-57 profile was used to measure PF, along with other QOL domains [25]. The PROMIS-57 profile includes 8-item short forms, which are valid and reliable for use in diverse clinical settings. The following QOL domains were evaluated as potential predictors of change in PF during the intervention: anxiety and depression (mental health), fatigue, pain, sleep disturbance, and sleep impairment (physical health), as well as participation in social roles/activities (social health) [26,27,28,29]. Surveys were scored utilizing the free Health Measures Scoring Service (https://www.assessmentcenter.net/ac_scoringservice; accessed on 3 July 2021). Cronbach’s alphas for these scales ranged from 0.85 to 0.97, except for sleep disturbance, which had an alpha of 0.24. Scores were normalized to the general population (mean = 50 points, SD = 10 points) to enhance the study findings, allowing for comparison.

Device-based measures of PA were assessed using a research-grade accelerometer/inclinometer, the activePAL3. Participants wore the activPAL3 monitor day and night for seven days at each of the three time points. This monitor provides accurate measurement of steps taken [30,31,32,33]. The number of minutes per day spent stepping at any intensity was also evaluated as a potential predictor of change in PF. Additionally, self-reported structured exercise was assessed using the Godin’s Leisure-Time Physical Activity Questionnaire [34,35]. Frequency (times per week) and average duration (in minutes) of exercise sessions that lasted a minimum of ten minutes were recorded based on intensity level: mild (e.g., light walk), moderate (e.g., brisk walking), and strenuous (e.g., run). Since this questionnaire just assessed time spent on structured exercise (moderate-to-strenuous activity), it differed from time spent stepping (all intensity levels) as assessed by the activePAL3 monitor.

### 2.4. Data Analysis

Baseline characteristics of the study participants are summarized with frequencies (percentages) or means (SDs) and ranges. Linear mixed effects models were used to evaluate potential associations of baseline QOL and physical activity, as well as changes in QOL and physical activity from baseline to the 6-month follow-up, with trends in PF from the six-month to nine-month follow-up periods. The models included a fixed effect for baseline values, changes in values from baseline to 6-month follow-up, time (six- vs. nine-month follow-up period), and baseline–time and change–time interactions. A subject-level random effect was included to account for correlation between repeated measurements obtained from the same individual over time. Models were adjusted for age and sex. Adding time since cancer diagnosis to the regression models did not appreciably alter the estimates, and was not included in the final model. Changes in the PF score (points) were examined per 5-point change in QOL scores and 30 min change in physical activity. *p* values < 0.05 were considered statistically significant. Due to the exploratory nature of this analysis, no corrections were made for multiple testing. Statistical analyses were conducted using SAS (version 9.4).

## 3. Results

The baseline characteristics of study participants are highlighted in Table 1. Average age at the time of enrollment was 68 years (range, 50–83). Most participants were female, non-Hispanic White, lived alone, and reported two or more comorbidities (83%), yet described their health as “good” or “very good to excellent” (83%). Breast (*n* = 11) and prostate (*n* = 6) represented the majority (57%) of cancer types. Participants averaged 98.6 stepping minutes per day (SD, 34.8) and self-reported 24.7 min per week (SD, 39.5) of moderate-intensity PA. On average, QOL measures of PF, anxiety, depression, and sleep impairment in cancer survivors who participated in this study were 1.3–3.4 points lower than the population average. Conversely, QOL measures of satisfaction with social roles/activities, fatigue, pain, and sleep disturbance were 0.1–2.7 points higher than the population average [36]. Note that for physical and social function, higher scores indicate better functioning, whereas for the remaining domains, higher scores indicate worse functioning.

Physical function at baseline was relatively high, with six out of eight questions averaging between four (able to do with little difficulty) and five (able to do without any difficulty). Scores were highest for being able to run errands and shop (4.67 out of 5) and being able to walk for at least 15 min (4.53 out of 5). Scores were lowest for activities in which their health limited their ability to do heavy work around the house such as scrubbing floors or lifting or moving heavy furniture (3.50 out of 5) or performing two hours of physical labor (3.77 out of 5). During the study, 47% of cancer survivors reported declines in their overall PF scores. While worse scores were reported for each of the eight items, the greatest declines were associated with being able to do heavy work around the house (34% reporting worse scores) and performing two hours of physical labor (38% reporting worse scores).

Worse depression, fatigue, pain, and sleep impairment at baseline were significantly associated with worse average PF scores across follow-up (2.3 to 4.9 points lower for every 5-point decrease in baseline QOL score; *p* values < 0.02) (Figure 1). Every 5-point increase in pain at baseline, or an increase in pain from baseline to the six-month follow-up, was associated with a significantly lower average PF over follow-up (by 4.9 and 3.0 points, respectively, *p* values < 0.001). Greater social participation and PA at baseline (both step and moderate-intensity minutes) were significantly associated with better average PF scores during the intervention (2.8 to 5.2 points higher for every 5-point increase in social participation or 30 min more of PA, *p* values ≤ 0.02), Figure 2. Worsening of depression, fatigue, and sleep impairment from baseline to the six-month follow-up was associated with an additional 2.1 to 2.9 point decrease in PF from the six-month to the nine-month follow-up (*p* values < 0.01), Figure 3).

## 4. Discussion

The current study is unique because it examined factors associated with changes in physical functioning among older cancer survivors who participated in a home vegetable gardening study conducted during the COVID-19 pandemic. Almost one-half (47%) of the study participants reported worse PF during the 9-month intervention than at the study baseline. Pain reported at baseline was the strongest predictor of decreased physical functioning, followed by fatigue, sleep impairment, and depression. Moreover, if these QOL measures worsened from baseline to the six-month follow-up, there was an association with an additional decrease in physical function over the subsequent three months of the study. Conversely, greater amounts of PA at baseline (both objectively measured and self-reported), as well as higher scores in satisfaction with social roles and activities reported at baseline or at the six-month follow-up, were associated with increased physical function.

Pain is very prevalent among cancer survivors. While rates are higher during treatment and for patients with metastatic disease, a recent meta-analysis reported a pain prevalence rate of 39.3% among post-treatment cancer survivors [37]. Pain is a well-known predictor of functional impairment and decline among older cancer survivors [38,39,40], so it is not surprising that it was associated with worse physical function during the study. This study did not assess whether pain management changed during the study, but some studies have reported reduced outpatient healthcare visits during the early stages of the pandemic [41]. While individuals with chronic pain may never become completely pain-free, achieving symptom tolerance can enable participation in other aspects of health.

Fatigue, depression, and sleep disturbance are also common symptoms in cancer survivors, and often co-occur [42,43,44]. Survivors often cite cancer-related fatigue (CRF), “a distressing, persistent, subjective sense of physical, emotional, and/or cognitive tiredness or exhaustion related to cancer or cancer treatment that is not proportional to recent PA and interferes with usual functioning [45]”. CRF can last for months or years following the completion of therapy [46]. Among cancer patients and survivors, between 30 and 50% report sleep disruption while 10–25% have reported depression [47,48]. In fact, depression in cancer survivors appears to be underdiagnosed and untreated [49]. Fatigue and depression also are well-known predictors of physical function decline among older cancer survivors [38,39,40]. There have been fewer studies examining the association between sleep disturbance and impaired physical functioning among cancer survivors, but evidence to date suggests that sleep disturbance is associated with problems with instrumental activities of daily living (IADLs) or poor self-reported physical function [50,51]. PA is recommended to cancer survivors given the numerous physical, mental, and social health and well-being benefits, including the attenuation or alleviation of these symptoms.

Just over half (53%) of cancer survivors maintained or improved their level of PF during the study. Maintenance or improvement was associated with greater amounts of PA at baseline or higher scores in satisfaction with social roles and activities (a measure of social participation [52]) reported at baseline or during the intervention. PA, especially moderate-to-vigorous intensity activity, is significantly associated with higher PF in adults [53]. Moreover, among older adults with comorbidities, evidence is emerging that light-intensity PA, such as gardening or leisurely walking, is also associated with better self-reported physical function and objectively measured PF (e.g., muscle strength, gait/mobility) [18,54,55,56]. Additionally, in older adults, including cancer survivors, social participation has been significantly associated with higher PF [57,58] or improvements in PF [59,60]. Higher social participation in both men and women also yields statistically significant increases in PA and reduced sedentary time [61].

Numerous health benefits have been associated with gardening [62,63,64,65,66,67]. A previous meta-analysis concluded that gardening activities were associated with better general health, mental health (reduced stress along with depression and anxiety symptoms), body mass index (BMI) scores, QOL, and sense of community [63]. Studies have also reported better socialization and social cohesion associated with gardening, perhaps due to collectively maintaining a garden, sharing produce, and discussing optimal growing strategies [62,65]. Gardening activities, classified as low-to-moderate-intensity PA [68,69,70], are associated with higher PA levels [63,66] and may serve as a gateway to additional types of PA, such as yard work, walking, or joining a gym [18,21]. Additionally, gardening activities can involve both upper and lower body functions such as kneeling, bending, lifting, reaching, and the use of small gardening tools. While most studies have focused on evaluating mental health, there is evidence that gardening, especially in older adults, results in better physical functioning [12]. In the Health and Retirement Study by the National Institute of Aging, gardeners aged 65–101 reported better balance and gait speed, and had fewer functional limitations compared to non-gardeners [66].

Notably, during the COVID-19 pandemic, interest in home gardening increased tremendously. Reports from several countries, including the United States, Canada, and Italy, indicated an increased demand for plants and other gardening products during the pandemic, in part from individuals who started growing vegetables or fruits at home for the first time [71,72,73]. In fact, the Master Gardener Program in Oregon experienced a 2806% increase in the number of people who signed up for online gardening courses [74]. This evidence points to an expanded global interest in gardening, an activity that confers health benefits such as improved well-being, PA, and lower perceived stress [11].

Compared to the majority of people who chose to create a home garden during the pandemic, the cancer survivors who participated in this intervention faced a more abrupt transition when COVID-19 started to spread throughout the United States. Study participants were adjusting to the newly implemented stay-at-home orders and other restrictions, social distancing, and closing of facilities where social and physical activities occur. This study did not assess the impact of the pandemic on QOL among study participants, most of whom, due to age and comorbidities, were at high risk for COVID-19 infection and poor outcomes.

### COVID-19 and Cancer Survivorship

The currently available literature suggests mixed results regarding the impact of the pandemic on older adults and older cancer survivors. A cross-sectional study highlights that compared to non-cancer participants, cancer patients reported higher self-efficacy in following national guidelines intended to slow the spread of COVID-19, higher levels of positive effects, and less stress [75]. On the other hand, in a longitudinal cohort study, both breast cancer survivors and age-matched controls reported incident sleep disturbance and increased loneliness [76]. In both survivors and control participants, increased loneliness was associated with significantly worsening depression, anxiety, and higher stress [77].

Maintaining social activity and PA during the pandemic has been a challenge for most individuals, especially those at high risk for infection. A cross-sectional survey of a global sample of cancer survivors reported a significant decrease in PA and QOL during the pandemic [78]. In addition, individuals with decreased PA and greater social isolation experienced worsened PF and fall outcomes [79]. In one study, cancer survivors reported decreased PA and prolonged periods of sitting [80]. Even individuals who were previously active experienced the aforementioned changes, in addition to weight gain [81]. These findings suggest deteriorating effects of the pandemic on cancer survivors and highlight the importance of promoting behaviors that foster social participation.

Limitations of this study include the small sample size and lack of a control group. Since the primary outcome explored the feasibility and acceptability of Southwest Harvest for Health, obtaining robust data on how the intervention impacted PF was not the priority. Since all participants in this study were enrolled in the gardening intervention, it cannot be concluded whether findings from this study are the result of gardening or confounding factors such as the pandemic and individual patient circumstances. It is possible that the association between increased PF and improved social participation and PA are a result of gardening efforts by participants over nine months. Vegetable gardening programs with a larger sample size and control group are needed to corroborate and expand upon initial findings. Additionally, this study did not collect data on recurrence or treatment started for cancer or other illnesses.

## 5. Conclusions

This initial Southwest Harvest for Health study demonstrated that worse pain, depression, fatigue, and sleep impairment before and/or during the first 6 months of the intervention were significantly associated with worse PF among older cancer survivors over a 9-month follow-up. The effects of the onset and early stages of the COVID-19 pandemic on PA and QOL measures in this high-risk population are unknown. However, encouraging social participation and PA through interventions such as home-based gardening may improve long-term health among older cancer survivors.

## Figures and Tables

**Figure 1 healthcare-10-01421-f001:**
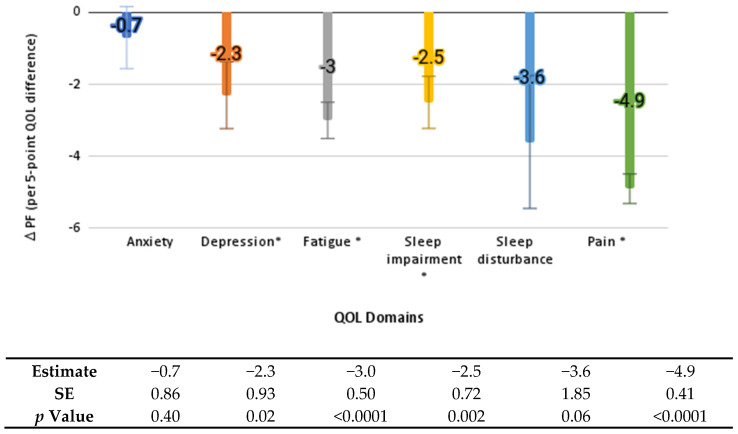
Change in physical function from baseline associated with baseline QOL domains. Error bars indicate standard error; * indicates statistical significance (*p* < 0.05). PF: physical function; QOL: quality of life; SE: standard error.

**Figure 2 healthcare-10-01421-f002:**
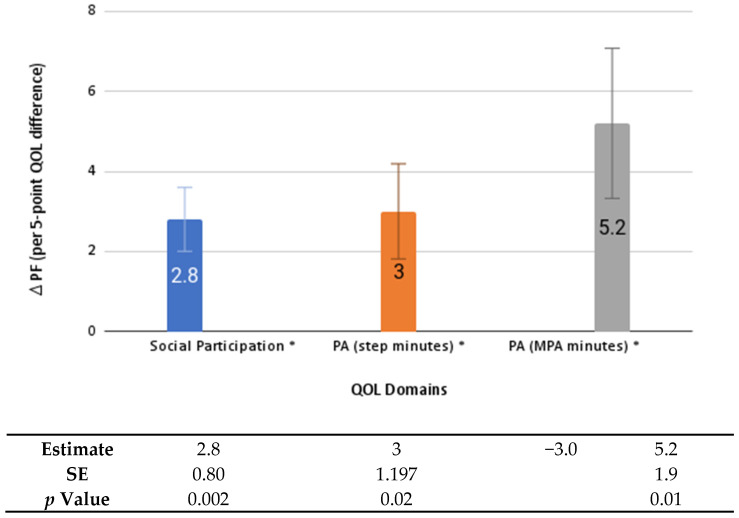
Change in physical function from baseline associated with baseline social participation and physical activity. Error bars indicate standard error; * indicates statistical significance (*p* < 0.05). PF: physical function; PA: physical activity; MPA: moderate physical activity; QOL: quality of life; SE: standard error.

**Figure 3 healthcare-10-01421-f003:**
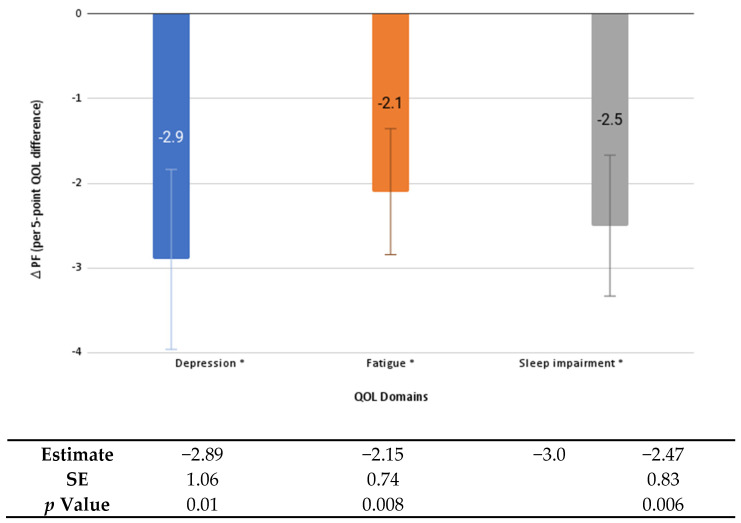
Changes in physical function from the six- to nine-month follow-up associated with five-point changes in QOL domains from baseline to the six-month follow up. Error bars indicate standard error; * indicates statistical significance (*p* < 0.05). PF: physical function; QOL: quality of life; SE: standard error.

**Table 1 healthcare-10-01421-t001:** Baseline characteristics of cancer survivors participating in the home-based, vegetable gardening study during the COVID-19 pandemic.

Characteristics	Frequency (%) or Mean (SD)
**Sociodemographics**	
Age	
50–64	9 (30%)
≥65	21 (70%)
Sex	
Female	21 (70%)
Male	9 (30%)
Race–ethnicity	
Non-Hispanic White	22 (73%)
Hispanic White	6 (20%)
Other	2 (7%)
Earned College Degree	
Yes	17 (57%)
No	13 (43%)
Income Group	
<USD 50,000	12 (40%)
≥USD 50,000	15 (50%)
Declined to answer	3 (10%)
**Health Characteristics**	
Self-reported general health	
Fair, poor	5 (17%)
Good	18 (60%)
Very good, excellent	7 (23%)
Living Arrangement	
Alone	13 (43%)
With Others	17 (57%)
Number of comorbidities	
0–1	5 (17%)
2–3	15 (50%)
≥4	10 (33%)
Type of cancer	
Breast	11 (37%)
Prostate	6 (20%)
Lung	4 (13%)
Other	9 (30%)
**Physical Activity/exercise**	
Objective physical activity (step minutes/day) ^1^	98.6 (34.8)
Self-reported exercise (minutes/week) ^2^	24.7 (39.5)
	Mean (SD) range
**Quality of Life** ^3^	
PF	47.9 (8.2) (29.7–60.1)
Satisfaction with social roles/activities	50.9 (9.1) (29.9–65.4)
Anxiety	48.1 (9.4) (37.1–67.0)
Depression	46.6 (8.4) (38.2–61.2)
Fatigue	50.2 (9.5) (33.1–63.2)
Pain	52.7 (8.4) (40.7–66.3)
Sleep disturbance	50.1 (5.0) (34.1–58.7)
Sleep impairment	48.7 (8.8) (35.9–60.6)

PF: physical function; SD: standard deviation. ^1^ Number of minutes/day spent stepping at any intensity; based on 7-day average with activPAL. ^2^ Self-reported moderate-intensity physical activity; number of minutes/week of moderate-to-vigorous intensity physical activity completed in minimum 10 min bouts. ^3^ For PF and social functioning, higher scores indicate better functioning. For the remaining domains, higher scores indicate worse functioning.

## Data Availability

Aggregate data may be available for research purposes upon reasonable request to the senior author (C.K.B.).

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
