# Peer review of "Association between Quality of Life and Physical Functioning in a Gardening Intervention for Cancer Survivors"

_healthcare, 2022, doi:10.3390/healthcare10081421_

Round 1
Reviewer 1 Report
This is a very interesting study on the affect of physical function on the quality of life of cancer survivors participating in a gardening intervention.
Although the effect of the gardening intervention was not examined separately, it seems that this is a beneficial activity for cancer survivors and people with comorbidities.
It would be helpful to add some information on the methodology of the study, namely how the 30 participants were selected to participate in the study, whether there were losses to follow up and whether the objectively measured physical activity with the accelerometer was compared with the reported activity in the same period. What were the exclusion criteria?
Moreover, did you account for time since cancer diagnosis or for periods undergoing treatment or other illness during the study period? These elements could greatly affect the level of physical function.
Author Response
Please find attached our response to reviewer 1.

Reviewer 2 Report
Age definition of older cancer survivors and age cut-off rational should be addressed in the introduction section.
P. 2, 48th line in introduction, sentences about the intervention procedure should be moved to methods section.
Why gardening intervention is beneficial especially for older cancer survivors? what aspect this intervention differs from other psychosocial intervention? Has this intervention done among cancer survivors before? why is it beneficial especially during the covid pandemic? Is it new to older cancer survivors? What prior studies said the associations between gardening intervention and the outcomes you measured in this study? especially, the association between gardening and social health? In general, the introduction section is too weak.
p. 2, 71st line, intervention was delivered and completed during pandemic? or before pandemic?
why did researchers include fruit and vegetable intake for eligibility criteria? need to address in introduction section or else where.
for eligibility criteria, is there no restrictions for cancer type, cancer stage, cancer treatment, time since diagnosis or treatment, or race/ethnicity? detailed information should be addressed even if there is no restriction for them.
p. 3, 101st line, should change "Patient Reported Outcomes Measurement Information System (PROMIS)-57 profile"
Should indicate the internal reliability for PROMIS-57 profiles in your data
Discussion section need to more focus on older cancer survivors, not general cancer survivors.
Author Response
Please find attached our response to reviewer 2.

Reviewer 3 Report
Thank you for the opportunity to learn about an interesting topic that is of paramount importance. In the introduction, the authors should include a definition of health-related quality of life, specify in the method whether the test used is general or specific, and supplement the literature with Chmielik, L.P., Mielnik-Niedzielska, G., Kasprzyk, A., Stankiewicz, T., Niedzielski, A. Physical and Psychosocial Concept Domains Related to Health-Related Quality of Life (HRQL) in 50 Girls and 52 Boys Between 5 and 18 Years Old in Poland Using the Parent-Reported 50-Item Child Health Questionnaire (CHQ-PF50),Medical Science Monitor, 2022, 28, e936801
Author Response
Please find attached our response to reviewer 3.

Round 2
Reviewer 2 Report
the manuscript has been sufficiently improved.
Author Response
We thank the reviewer for their comment that this manuscript has been sufficiently improved.